# Plasma-Functionalised Dressings for Enhanced Wound Healing

**DOI:** 10.3390/ijms24010797

**Published:** 2023-01-02

**Authors:** Xanthe L. Strudwick, Jason D. Whittle, Allison J. Cowin, Louise E. Smith

**Affiliations:** 1Future Industries Institute, University of South Australia, Mawson Lakes, SA 5095, Australia; 2UniSA STEM, University of South Australia, Mawson Lakes, SA 5095, Australia

**Keywords:** plasma polymerization, functionalization, dressing, healing, wound, closure, re-epithelialization, collagen

## Abstract

Fundamental knowledge about cell–surface interactions can be applied in the development of wound dressings and scaffolds to encourage wounds to heal. As surfaces produced with acid-functionalised monomers encourage keratinocyte adhesion, proliferation and migration, whilst amine functionalisation enhances fibroblast proliferation and migration in vitro, standard care wound dressings were plasma-coated with either acrylic acid or allylamine and applied to 6 mm excisional wounds on the backs of mice to test their effectiveness in vivo. At day 3, the rate of wound healing was increased in mice treated with dressings that were plasma-coated with allylamine compared to uncoated dressings, with a significantly reduced wound area. However, healing may be impaired following prolonged treatment with allylamine-functionalised dressings, with delayed re-epithelialisation and increased cellularisation of the wound site at later timepoints. Acrylic acid functionalisation, however, offered no early improvement in wound healing, but wounds treated with these dressings displayed increased collagen deposition at day 7 post wounding. These results suggest that plasma polymerisation may allow for the development of new dressings which can enhance wound closure by directing cell behaviour, but that the application of these dressings may require a timed approach to enhance specific phases of the wound healing response.

## 1. Introduction

Despite wound care costing Australia approximately AUD 2.85 billion per annum, it still receives little attention and is considered to be a silent epidemic [1]. Chronic wounds, in particular, take greater than 4 weeks to heal, requiring constant dressing applications to provide protection from bacterial infection, and thus present an increased burden to both the patient and community at large [2,3]. The use of dressings in the routine treatment of chronic wounds provides a therapeutic window for the development of wound-care dressings which may be able to drive the wound healing response. Recent advances have seen the arrival of hydrogel dressings and antibacterial smart dressings to the wound-care market which absorb wound exudate whilst providing hydration and protection and may also reduce the bacterial burden of the wound, but chronic wounds remain a significant global clinical challenge to manage [4,5,6,7].

Fundamental knowledge about the interactions of cells with the surfaces upon which they grow can be applied in the development of wound dressings and scaffolds to encourage chronic wounds to heal [8,9]. Plasma polymerisation enables functional groups to be deposited onto 2D or 3D substrates, in a solvent-free, xenobiotic-free process which can be tuned to produce surface chemistries that can be used to direct specific cell behaviours [10,11,12,13,14,15,16]. Acrylic acid and allylamine are two commonly used monomers used in plasma polymerization for biofunctionalization and biological applications [8,17,18,19,20,21,22,23,24,25]. Acrylic acid plasma polymers have an inherent negative charge when in an aqueous environment, whilst allylamine plasma polymers have a neutral to positive charge. This range allows for the adsorption of a range of proteins to the surface and the ability to select for specific proteins, which in turn allows for the tuning of cell behaviour. We have previously shown that keratinocytes and endothelial cells, i.e., cells that grow on a basement membrane, prefer to be cultured on an acrylic acid plasma polymer and will struggle to attach to an allylamine plasma polymer surface. Conversely, fibroblasts and other mesenchymal cells, whilst able to attach to most plasma polymer surfaces, prefer to attach to an allylamine or positively charged surface [8]. In addition to utilising single monomer species such as acrylic acid or allylamine, copolymerisation of multiple monomers can be performed simultaneously and even applied as gradients across a surface when deposited in a stepwise fashion or in combination with masking to alter or disrupt the diffusion of the monomer within the plasma reactor [18,26,27,28]. For a review on the mechanisms of acrylic acid plasma polymer deposition, we strongly recommend [29]. Daunton et al. and Ryssy et al. also investigated the retention of amine functional groups during plasma polymerization [30,31].

We have previously used plasma polymerisation to functionalise tissue culture plastic to direct cell behaviour. These initial in vitro studies determined that the culture of keratinocytes and fibroblasts was best supported by tissue culture plastic that was coated with 100% functional monomers, rather than those diluted with an octadiene comonomer. In particular, keratinocyte viability was highest, and the cells formed colonies and exhibited lamellipodia, migrating fastest on 100% acrylic acid-functionalised surfaces, which have approximately 20% COOH/R (carboxyl or ester) functionality. Conversely, we found that fibroblasts grown on the allylamine surfaces with approximately 15% primary amine functionality exhibited the highest viability, had improved morphology, were better spread and flatter with some filopodia present, and migrated significantly faster on this surface [8]. Utilising this same plasma polymerisation technique, we have also functionalised synthetic electrospun poly-L-lactic acid fibres with an allylamine to stimulate human dermal fibroblasts in vitro to migrate faster into the scaffold, adhere and rapidly proliferate to populate the scaffold, and subsequently deposit significantly more collagen I on the scaffolds than untreated controls, identifying this scaffold as a suitable dermal replacement for deep wounds with impaired healing [9]. In this current study, we aimed to utilise plasma polymerisation to functionalise standard wound care dressings with acrylic acid or allylamine and develop dressings which may enhance wound healing and lead to faster re-epithelialisation and improved collagen deposition. These functional dressings may then provide an avenue for therapeutic wound care to reduce the burden of chronic wounds.

## 2. Results

### 2.1. Production and Administration of Plasma-Functionalised Dressings

Plasma polymer-coated dressings were produced, and the surface chemistry was analysed by X-ray photoelectron spectroscopy (XPS). Spectra (Figure 1 with elemental peak fits summarised in Appendix A) revealed the presence of approximately 20% carboxyl functionality and 20% alcohol/amine functionality on the acrylic acid- and allylamine-functionalised surfaces, respectively. Uncoated Melolin contains approximately 12% carboxyl and 16% alcohol functionality, as would be expected of a polyester. Uncoated and acrylic acid- or allylamine-functionalised dressings were applied to murine excisional wounds and left to heal for 3, 5 and 7 days. Previous work [8] has shown that these deposited acrylic acid and allylamine plasma polymer films have average thicknesses of 26.84 nm and 30.01 nm, measured by atomic force microscopy. When placed into an aqueous environment, the acrylic acid film largely dissolves and leaves a highly negatively charged acid film approximately 5 nm thick. Conversely, the allylamine film swells slightly, producing a neutrally charged film with an average thickness of 31 nm [8]. Additionally, both the acrylic acid and allylamine films had similar roughness when dry and deposited on smooth surfaces, i.e., silicon wafer to enable accurate measurement (RMS (Root Mean Square) = 0.34 and 0.31, respectively). The film roughness increased when wet (RMS = 0.48 and 0.91, respectively) [8]. It is important to note that these measurements were taken after films were deposited onto silicon wafers, which are hard, smooth substrates that allows for accurate measurements. Melolin is a cast polyester film and is therefore relatively soft and not smooth. Therefore, there will be subtle differences in the thickness of the films due to differences in the bonding of the plasma polymer to a polymeric substrate rather than a ceramic or glass substrate. Additionally, regarding the inherent texture of the Melolin, which is visible to the naked eye and will therefore be of micron, even millimeter-scale features will mask any changes in the film thickness and/or roughness due to swelling and/or dissolution.

### 2.2. Allylamine Functionalisation Accelerates Early Wound Closure in Murine Excisional Wounds

Morphometric analysis of digital images of the wounds (Figure 2A) revealed a significant decrease in the area of the wounds (Figure 2B) which were treated with allylamine-functionalised dressings on day 3 post wounding when compared to uncoated control dressings. Wounds treated with acrylic acid-functionalised dressings were significantly larger on day 5 compared to wounds treated with uncoated control dressings. By day 7, no significant differences between groups were observed, with all wounds healing to around 40% of their initial wound area.

### 2.3. Re-Epithelialisation May Be Delayed by Treatment with Allylamine-Functionalised Dressings

Histological analysis of the wounds (Figure 3A) showed that while there were no significant differences in the length of the wounds across the midline (Figure 3B) on day 3 or 7, the wounds of mice treated with allylamine-functionalised dressings appeared significantly wider on day 5 post wounding. Re-epithelialisation (Figure 3C) was also reduced in these day-5 wounds treated with allylamine-functionalised dressings compared to uncoated controls. This trend continued at day 7 but was no longer significant. A nonsignificant increase in re-epithelialisation was also observed in day-3 wounds treated with acrylic acid-functionalised surfaces, but this trend was not observed at the later timepoints.

### 2.4. Plasma-Functionalised Dressings Increase Collagen Deposition and Cellular Infiltration

Histological sections were then stained with Masson’s trichrome (Figure 4A), which selectively stains tissue components according to density of the tissue, such that collagen will appear blue/green, the cytoplasm of cells will appear red and nuclei blue/black. Semiquantification of collagen deposition (Figure 4B) revealed a significantly higher score for collagen/green staining in sections from day-7 wounds which had been treated with acrylic acid-functionalised dressings compared to uncoated control dressings. No significant differences were observed at earlier time points or in wounds treated with allylamine-functionalised dressings.

Quantitative measurement of the RGB (red, blue, green) images of day-7 sections showed a decrease in the amplitude of blue to red intensity from the control intensity of 0.366 ± 0.049 down to 0.278 ± 0.074 in the allylamine-functionalised dressing cohort. Acrylic acid, which achieved the greatest collagen deposition score by semiquantitative analysis, also displayed the highest value, at 0.456 ± 0.066. Semiquantification of the red staining, which represents the cellular infiltrate within the wounds, revealed that wounds treated with allylamine-functionalised dressings had significantly increased cellularisation of the wound site on day 7 post wounding (Figure 4C).

Analysis of the wound tissue by immunofluorescent detection of inflammatory (myeloid-related protein (MRP)-14 positive) or endothelial (platelet endothelial cell adhesion molecule, also known as cluster of differentiation 31 (CD31) positive) cells (Figure 5A) revealed that this cellular infiltrate was not inflammatory cells, with all three groups displaying similarly low numbers of MRP-14 positive cells at the day 7 timepoint, which is when the inflammatory phase is usually resolved (Figure 5B). Interestingly, the number of CD31 positive endothelial cells (Figure 5C) in the allylamine-treated group appeared to be elevated compared to the acrylic acid and uncoated groups; however, this trend did not reach significance by day 7.

## 3. Discussion

With the ever-increasing prevalence of chronic wounds and a predicted global advanced wound-care market of AUD 18.7 billion by 2027, there are calls for increasing attention to be placed on the development of new approaches to stimulate the healing [33]. As chronic wounds require covering with wound dressing to prevent bacterial infection [34], they present an ideal opportunity to direct cellular behaviour within the underlying wound tissue through plasma functionalisation of the dressing surface.

Our initial in vitro studies showed that while the allylamine-functionalised surface improved fibroblast migration, keratinocytes migrated slower on allylamine films. Conversely, where the acrylic acid surfaces supported keratinocyte migration, these proved to be poor at supporting fibroblast migration [8]. Therefore, Melolin dressings were functionalised with allylamine or acrylic acid, and their effects on wound healing rates were compared to the standard uncoated dressing. While a significant acceleration in early wound healing was observed when dressed following allylamine functionalisation, this effect was transient and in fact inhibited re-epithelialisation of the wound. It is likely that the opposing migratory effects of allylamine-functionalised surfaces seen in vitro may have led to the delay in re-epithelialisation observed in the wounds covered with allylamine-functionalised dressings at day 5 post wounding and reduced the rate of healing overall. It appears that the initial positive effect of these dressings upon early wound closure, potentially though increased migration of fibroblasts into the wound site and more rapid provisional matrix deposition, is later offset through the prevention of keratinocyte migration and formation of a neoepidermis.

Moreover, fibroblasts had previously been shown to deposit more collagen in vitro when cultured upon allylamine-functionalised surfaces [9,35,36]. Interestingly, we have seen here that the collagen deposition was greatest in wounds treated with acrylic acid-functionalised surfaces, with allylamine surfaces no different to control. Deposition and remodelling of the extracellular matrix is facilitated by fibroblasts with a less migratory, and more contractile, phenotype [37]. It may be that the reduced migration of fibroblasts cultured on the acrylic acid surfaces in vitro [8] is indicative of this and may explain the increased collagen deposition within the wounds covered with acrylic acid-functionalized dressings; however, further investigations into this phenomenon will be carried out in future studies to determine this. It had been expected that functionalisation with acrylic acid would support healing processes by promoting the growth and migration of keratinocytes for re-epithelialisation and endothelial cells for angiogenesis, whereas those functionalised with allylamine would support fibroblasts which are responsible for collagen deposition and dermal regeneration [8]. Our initial in vitro studies suggested that endothelial cells would react better to the acrylic acid-functionalised dressings, growing significantly better on the acrylic acid series of surfaces than the allylamine ones and showing increased viability on acrylic acid surfaces compared to standard tissue culture plastic [8].

Interestingly, the wounds treated with allylamine-functionalised dressings exhibited significantly increased cellular infiltrate on day 7 post wounding. Our immunohistochemical analysis revealed that this increased cellularisation was not due to an increased immune response, with similar numbers of MRP-14 positive immune cells (which include monocytes, pro-inflammatory macrophages, neutrophils and granulocytes [38,39]) within the wounds of mice treated with allylamine-functionalised dressings compared to control dressings. Using CD31 staining, which detects endothelial cells to visualise blood vessels within the skin [40], it was revealed that the increased cells observed may in fact have been endothelial cells, suggesting that the allylamine-functionalised dressing might improve blood vessel formation within the healing wound.

In this study, we observed a similar trend to our in vitro work that indicated that what works for one cell type does not always work well for all (Smith et al. 2016 saw that keratinocytes and endothelial cells responded similarly but opposite to fibroblasts [8]); however, the differences observed between the in vitro and in vivo models demonstrate the added complexity of interactions between cell types and the extracellular environment within the healing wound. Taken together, the results suggest that the application of allylamine-functionalised dressings to excisional wounds could be most beneficial when restricted to the initial phases of wound healing to stimulate rapid granulation tissue formation and then removed so as not impede re-epithelialisation and overall wound closure. Alternatively, the application of allylamine-functionalised dressings may be more useful for treating chronic, nonhealing wounds, which would benefit by stimulation of angiogenesis [41], and dressings functionalized by acrylic acid are better for wounds in elderly patients at greater risk of wound dehiscence, where increased collagen deposition may be beneficial [42]. However, the utility of the dressings in these circumstances is not investigated in the current study. Nevertheless, this study has shown that the application of plasma-functionalised dressings can impact the healing profile of wounds and modulate cellular behaviour within the wound site.

## 4. Materials and Methods

### 4.1. Plasma Polymerisation of Dressings

Acrylic acid and allylamine were purchased from Sigma-Aldrich (Castle Hill, New South Wales, Australia) at > 99% purity and used as received. Sterile Melolin nonadherent dressings (Smith & Nephew, North Ryde, New South Wales, Australia) were placed into a custom-built plasma reactor, described previously [35], comprising a cylindrical stainless steel vacuum vessel with a diameter of 30 cm and a volume of approximately 20 L. Acrylic acid and allylamine films were deposited onto the surface of the Melolin dressings as previously described [8]. Briefly, a total monomer flow rate of ca. 4 sccm (∼1.4 × 10^−2^ mbar) was used for depositing acrylic acid films for 20 min after ignition of the plasma using a radio frequency generator at 13.56 MHz and a power of 3 W, whilst a total monomer flow rate of ca. 5 sccm (∼1.9 × 10^−2^ mbar) was used for deposition of allylamine films for 35 min after ignition of the plasma using a radio frequency generator at 13.56 MHz and a power of 5 W. A schematic of the plasma reactor is included below (Figure 6).

### 4.2. Surface Characterisation

Surface chemistry of uncoated and plasma-functionalised dressings was characterized by X-ray photoelectron spectroscopy performed using a SPECS SAGE XPS system with a Phoibos 150 hemispherical analyser and MCD-9 detector (SPECS Surface Nano Analysis GmbH, Berlin, Germany). All results were obtained using a non-monochromated Mg Kα radiation source (hν: 1253.6 eV) operated at 10 kV and 20 mA (200 W). The analysed area was circular and 5 mm in diameter. Survey spectra (0−1000 eV binding energy) were collected at a pass energy of 100 eV, with a resolution of 0.5 eV. To determine the chemical functionalities, high-resolution spectra of the C 1s core level peak were collected at a pass energy of 20 eV, with a resolution of 0.1 eV. Resulting spectra were analysed using CasaXPS software (Neal Fairley, Devon, UK). A linear background was used, while the full width at half-maximum was set to be between 1.2 and 1.5 eV (and fixed for all peaks in any specific fit). The area under the photoelectron peaks was used to calculate the atomic percentage using manufacturer-supplied relative sensitivity factors. The peak fits used 70% Gaussian/30% Lorentzian peak shapes. Amine plasma polymer fitted according to Robinson et al. 2014 [32], with C1 at 285 eV (C–C), C2 at 286.4 eV (C=N, C–OR and C–N) and C3 at 287.5 eV (C=O and C(=O)OH/R). All binding energies were referenced to the aliphatic C−C/C−H peak at 285.0 eV.

### 4.3. Excisional Wound Model

All animal procedures were approved by the Women’s and Children’s Health Network Animal Ethics Committee (AE953/9/16) and carried out in accordance with the Australian code of practice for the care and use of animals for scientific purposes. Two 6 mm circular, full-thickness excisions were created 0.5 cm either side of midline on the shaved and depilated dorsum of 8-week-old female Balb/C mice (*n* = 6/group) and immediately covered with 1 cm × 1 cm squares of Melolin wound dressings, either uncoated or plasma-functionalised with acrylic acid or allylamine and secured with Tegaderm adhesive dressing. Dressings were left in place for the duration of the trial and reapplied as required if removed by the animal. Digital images were taken of the wounds on day 0, 3, 5 and 7 post wounding, the animals were humanely killed by CO_2_ with cervical dislocation on day 3, 5 or 7 post wounding and skin samples were collected for postmortem analysis.

### 4.4. Wound Healing Analysis

Morphometric analysis of the wounds was performed on digital images using Image Pro Plus software to determine the area of the wounds. Wounds and surrounding intact skin were surgically excised to the fascia, bisected along the transverse plane and fixed overnight in 10% neutral buffered formalin prior to embedding in paraffin. The 4 µm sections were cut from paraffin-embedded formalin fixed-tissue samples and subjected to hematoxylin- and eosin- (H and E) staining to visualise the skin structures for histological assessment of wound closure over time. Histological measurements were performed on 4× magnification images in Image Pro Plus to determine the wound length (the distance between the first hair follicles either side of the wound), and re-epithelialisation was calculated as the percentage of neoepidermis within the total wound length.

### 4.5. Collagen Analysis

Collagen deposition was analysed in histological sections using Masson’s Trichrome. Stitched bright field images of the entire Masson’s trichrome-stained sections were taken using a 10× objective on a one-deck, fully motorized IX83 inverted imaging system with a DP80 camera (Olympus, Tokyo, Japan). Semiquantitative scoring of collagen deposition was performed using a scale of 1 (very little blue/green staining compared to surrounding intact skin) up to 5 (similar to surrounding intact skin). ImageJ (National Institutes of Health, Bethesda, Maryland, USA) was used for the quantitative morphometric analysis of Masson’s trichrome slides using the Macro described by Kennedy et al. 2006 [43], which allocates each pixel with an intensity of blue greater than 120% of the intensity of red and a grey scale amplitude of 1, leaving all other pixels an amplitude of 0 and calculating the amplitude of blue staining as a fraction of total area of pixels. Cellularisation was also assessed semiquantitatively from Masson’s trichrome-stained sections, using a scale from 1 (very few cells present) up to 5 (greatly increased cells present).

### 4.6. Immunofluorescent Analysis

Serial sections were also assessed for inflammatory cell infiltrate (Santa Cruz, Dallas, Texas, USA goat anti-Calgranulin B/MRP-14: sc-8114, 100 µg/mL @ 1:200) and endothelial cell presence (Abcam, Cambridge, UK rabbit anti-CD31: ab28364, 100 µg/mL @ 1:200) within the wound site by immunohistochemistry following antigen retrieval by heat in the Decloaking Chamber (Biocare Medical Inc, Pachecho, California, USA) at 90 °C for 10 min, followed by enzymatic digestion with 0.0625 g Trypsin from Porcine Pancreas (Sigma-Aldrich, Castle Hill, New South Wales, Australia) in 250 mL 37 °C PBS for 3 min. Nonspecific antibody binding was blocked with 3% normal goat or horse serum in PBS for 30 min at room temperature, followed by overnight primary antibody incubation in a moist airtight box at 4 °C. After removal of excess primary antibody, incubation with the appropriate species-specific antibody (Invitrogen, Waltham, Massachussetts, USA Alexa Fluor donkey antigoat 568: A11057, 2 mg/mL @ 1:500 or Alexa Fluor goat antirabbit 565: A11011, 2 mg/mL @ 1:500) was performed at room temperature in a moist box and protected from light for 1 hr, followed by nuclear counter stain with 2 µg/mL 4′,6-Diamidino-2-phenylindole dihydrochloride (Sigma-Aldrich, Castle Hill, New South Wales, Australia) for 2 min. Immunofluorescence was detected using an Olympus IX83 microscope and DP80 camera (Olympus, Tokyo, Japan). Images were captured using a 20× Objective and with 0.5× gain. Images were then analysed using the cellSENS Dimension Microscope Imaging Software v 2.3 (Olympus, Tokyo, Japan) to measure either positive cells per area or mean grey intensity, which is automatically calculated as intensity per area.

### 4.7. Statistical Analysis

Statistical analysis was performed using GraphPad Prism 6 software (GraphPad Software, San Deigo, California, USA). Data were analysed using a one-way ANOVA with Dunnett’s or Student’s T post-tests. A *p*-value of less than 0.05 was considered significant.

## 5. Conclusions

Plasma functionalisation of wound dressings with acrylic acid or allylamine altered the healing profile of mouse wounds. These results suggest that plasma polymerisation may allow for the development of new dressings which can enhance wound closure by directing cell behaviour, but that the application of these dressings may require a timed approach to enhance specific phases of the wound healing response.

## Figures and Tables

**Figure 1 ijms-24-00797-f001:**
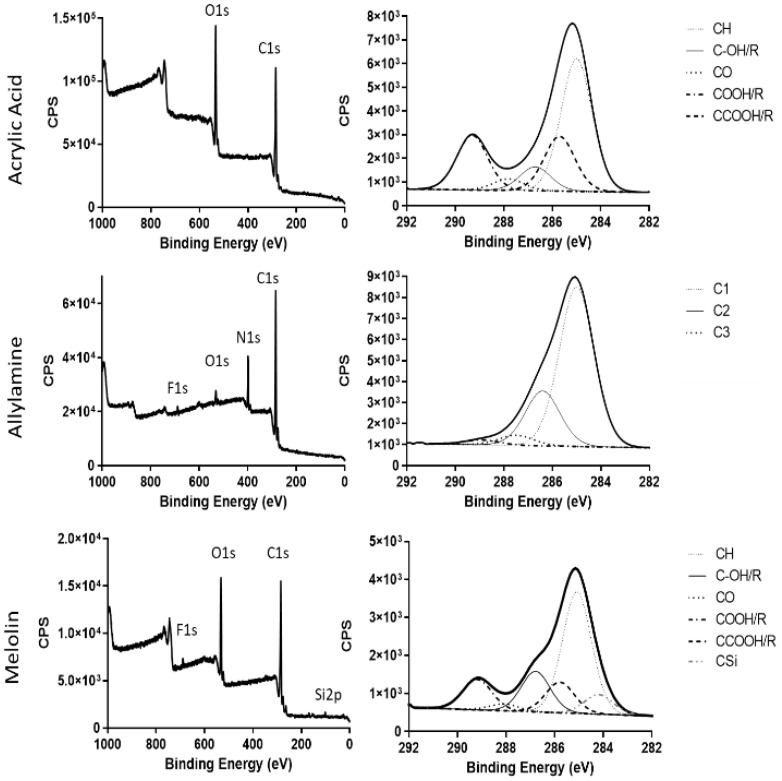
Survey and corresponding C1s X-ray Photoelectric Spectroscopy XPS spectra from uncoated Melolin and Melolin coated with either an acrylic acid or allylamine plasma polymer. Amine plasma polymer fitted according to Robinson et al. [32], whereby C1 at 285 eV (C–C), C2 at 286.4 eV (C=N, C–OR and C–N) and C3 at 287.5 eV (C=O and C(=O)OH/R).

**Figure 2 ijms-24-00797-f002:**
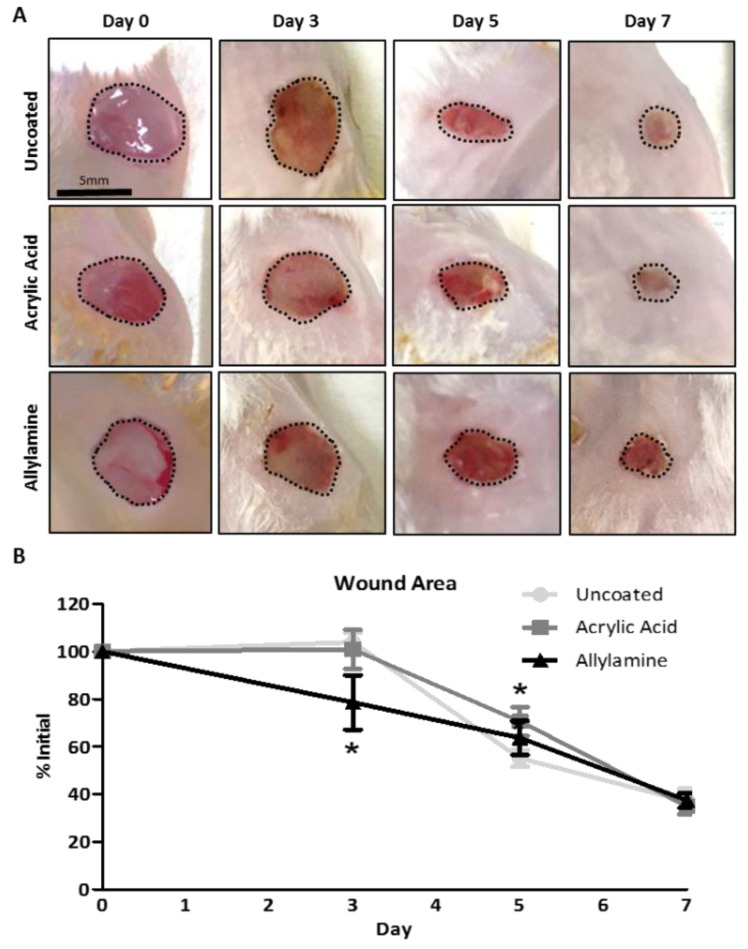
Treatment with allylamine but not acrylic acid-functionalised dressings improves early wound closure. (**A**) Wound area (dashed line) was measured from digital images of excisional wounds treated with acrylic acid or allylamine plasma-functionalised dressings on 0-, 3-, 5- and 7-days post wounding. (**B**) Wound area, expressed as a percentage of initial wound area, was significantly reduced in the allylamine group on day 3 but increased in the acrylic acid group on day 5. *n* = 6. Mean +/− SEM. * *p* < 0.05 vs. uncoated dressing.

**Figure 3 ijms-24-00797-f003:**
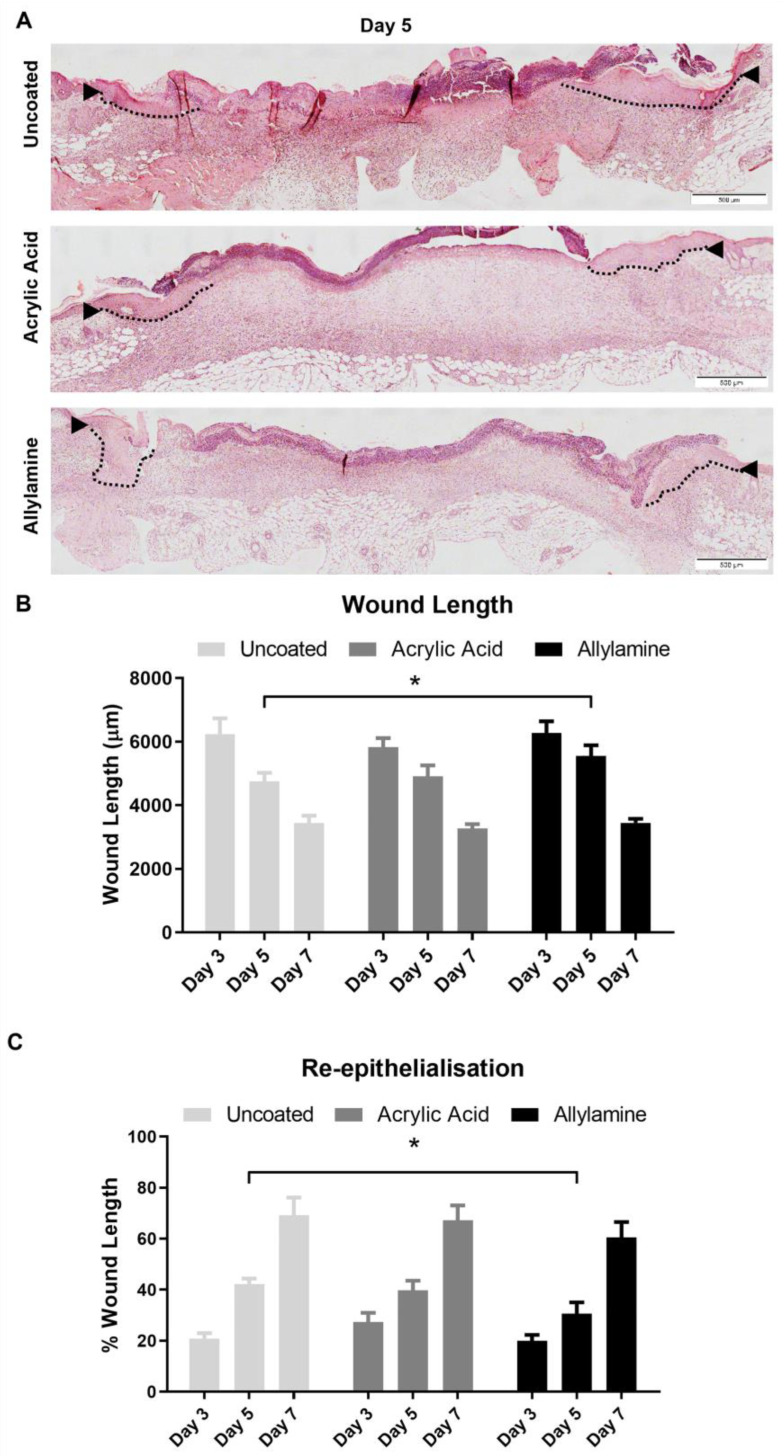
Treatment with allylamine-functionalised dressings impedes re-epithelialisation. (**A**) Haematoxylin- and eosin-stained 4 µm paraffin-embedded sections of wounds treated with uncoated and acrylic acid or allylamine-functionalised dressings were assessed for the wound length (arrow heads) and neoepidermis (dashed lines). Scale bar 500 µm. (**B**) Wound length was calculated as the distance between the first hair follicles either side of the wound. (**C**) Re-epithelialisation was calculated as the percent of wound length covered by neoepidermis. *n* = 6. Mean +/− SEM. * *p* < 0.05 vs. uncoated dressing.

**Figure 4 ijms-24-00797-f004:**
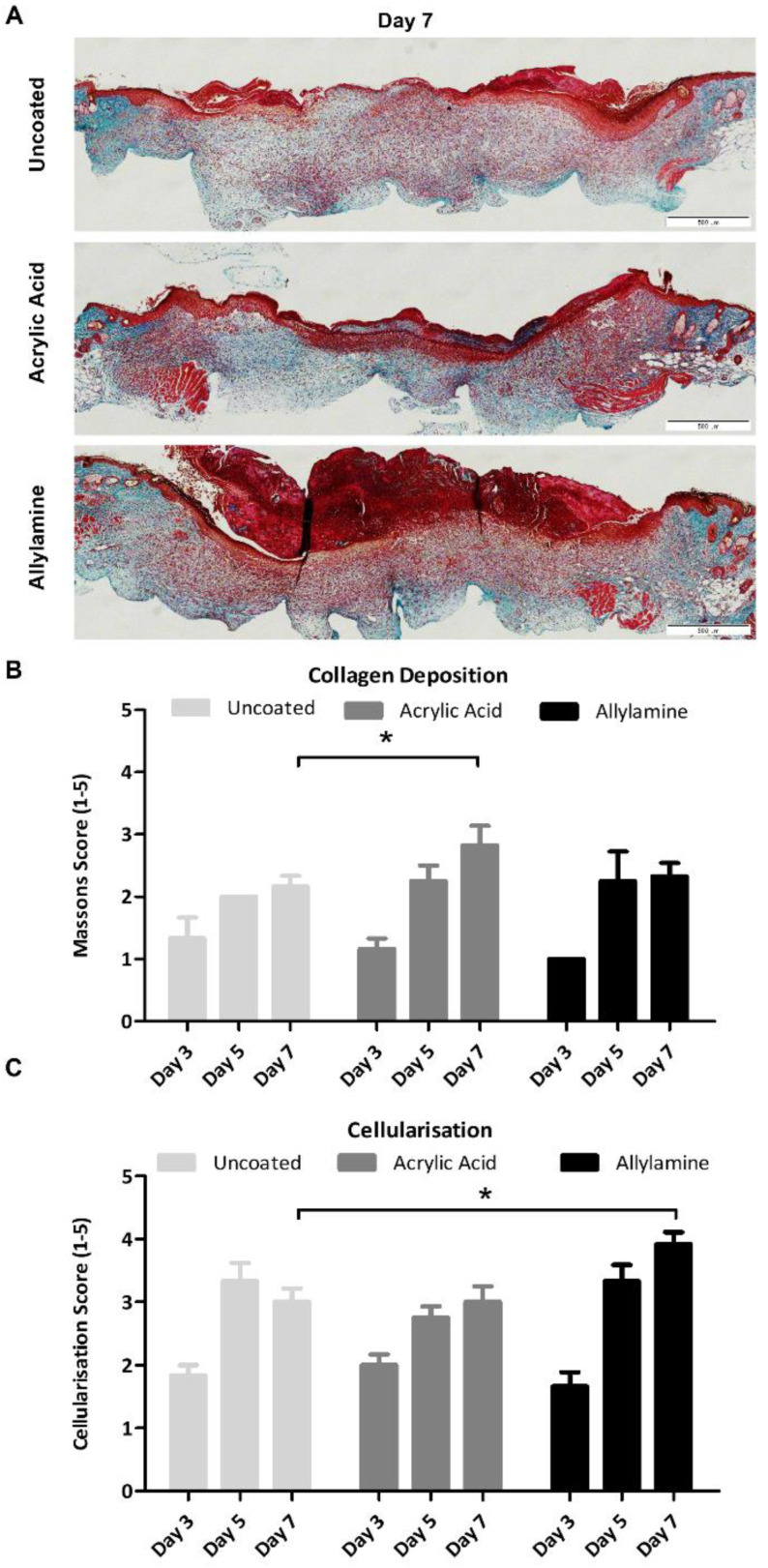
Collagen deposition is increased following treatment with acrylic acid but not allylamine-functionalised dressings. (**A**) Masson’s trichrome staining was performed on 4 µm paraffin-embedded sections of wounds treated with uncoated and acrylic acid or allylamine-functionalised dressings to visualise collagen (blue/green), cell cytoplasm (red) and nuclei (blue/black). Semiquantification of (**B**) collagen deposition (relative abundance of blue/green collagen staining) and (**C**) cellularisation (relative abundance of red cellular staining) compared to adjacent unwounded tissue was performed with a score given from 1–5. *n* = 6. Mean +/− SEM. * *p* < 0.05 vs. uncoated dressing.

**Figure 5 ijms-24-00797-f005:**
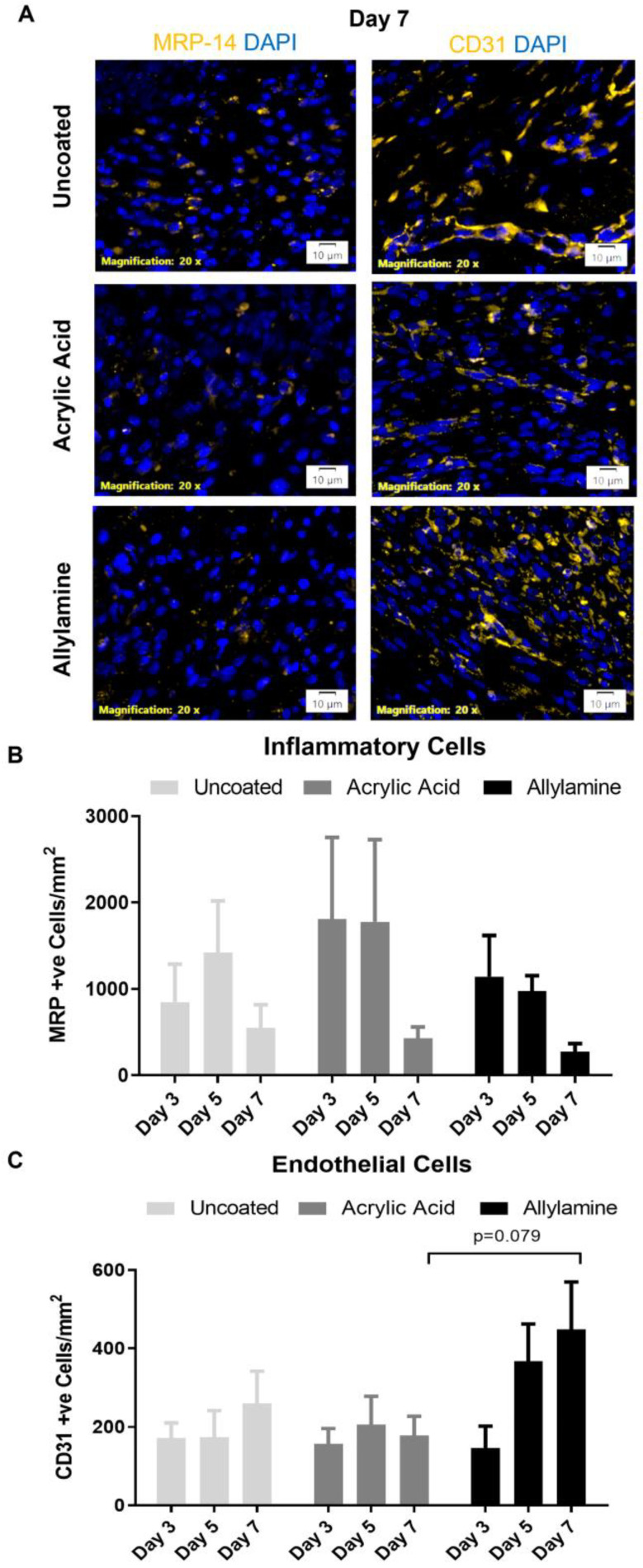
Treatment with allylamine-functionalised dressings may increase endothelial cells’ numbers within the wound. (**A**) Inflammatory cell marker myeloid related protein (MRP)-14 and platelet endothelial cell adhesion molecule, also known as cluster of differentiation 31 (CD31) immunofluorescent staining was performed on 4 µm paraffin-embedded sections of wounds treated with uncoated and acrylic acid or allylamine-functionalised dressings to visualise positive cells (orange) and nuclei (DAPI blue). Quantification of (**B**) MRP positive inflammatory cells and (**C**) CD31 positive endothelial cells within the wound was performed. *n* = 6. Mean +/− SEM.

**Figure 6 ijms-24-00797-f006:**
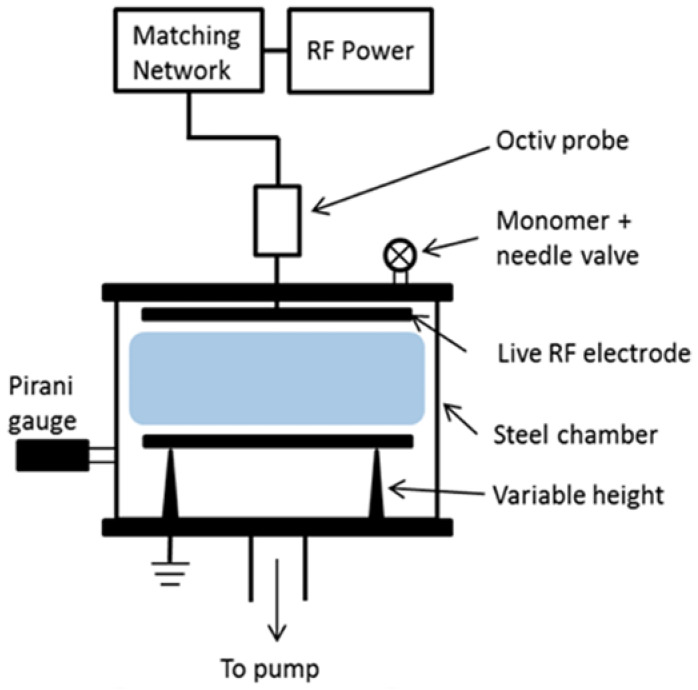
Schematic representation of plasma reactor used for depositing plasma polymers (reprinted (adapted) with permission from [35]). Copyright 2013 American Chemical Society.

## Data Availability

Data are contained within the article.

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
