# Peer review of "Plasma-Functionalised Dressings for Enhanced Wound Healing"

_ijms, 2023, doi:10.3390/ijms24010797_

Round 1

Reviewer 1 Report

In this manuscript, the authors examined the role of plasma coated allyamine and acrylic acid for the function of wound healing.  In their work, they claimed that stages of wound healing for both materials are time dependent as they had shown that at various time points, reepithelization occurred differently for both allyamine and acrylic acid.  On the basis of their observation, they suggested that a timed approach should be applied when performing wound healing although they did not elaborate on how this would be achieved.  Overall, the reviewer finds this work to be well-written and presented although was slightly hard-pressed to establish and arrive to the claims by the authors.  For example, the cellularization differences between allylamine and acrylic acid was rather negligible for day 5 and it was not statistically significant for collagen deposition for day 7 in the reviewer’s opinion.  Furthermore, the overall results between uncoated controls and the plasma deposited surface had not be overly convincing.  The error bars for figure 5 was also a major concern for the reviewer when substantiating some of these claims.  Hence, despite the well-written nature and good quality presentation, the overall results had been rather underwhelming and coupled by the fact that there is rather vague how a “timed phased” could be applied to potential wound healing work, the reviewer felt that this work remains rather unpolished.

Author Response

We thank the reviewer for the time taken to assess the submission. We agree with the reviewer regarding the day 5 cellularization and do not report on any difference between the allylamine and acrylic acid functionalised surfaces in the manuscript in Section 2.4. Lines 151-154 and Figure 4C indicate that the significant difference in cellularisation occurs between the allylamine and uncoated dressings. ”Semi- quantification of the red staining which represents the cellular infiltrate within the wounds, revealed that wounds treated with allylamine functionalised dressings had significantly increased cellularisation of the wound site on day 7 post wounding (Figure 4C).” Similarly, lines 141-146 and Figure 4B report on the significant difference in collagen between the uncoated and acrylic acid dressings and does not claim any significant difference between the acrylic acid and allylamine functionalised dressings. “Semi-quantification of collagen deposition (Figure 4B) revealed a significantly higher score for collagen/green staining in sections from day 7 wounds which had been treated with acrylic acid functionalised dressings compared to uncoated control dressings. No significant differences were observed at earlier time points, or in wounds treated with allylamine functionalised dressings.” While there is variability in number of MRP14 +ve inflammatory cells observed within the wounds, particularly at days 3 and 5 in the acrylic acid treated group, the results clearly demonstrate that the increased cellularisation observed on day 7 in the allylamine group is not attributed to elevated inflammatory cell infiltration, but likely due to the increased numbers of CD31 +ve endothelial cells, as discussed in the manuscript lines 155-163. “Analysis of the wound tissue by immunofluorescent detection of inflammatory (myeloid-related protein (MRP)-14 +ve) or endothelial (Platelet endothelial cell adhesion mol-ecule also known as cluster of differentiation 31 (CD31) +ve) cells (Figure 5A) revealed that this cellular infiltrate was not inflammatory cells with all three groups displaying similarly low numbers of MRP-14 +ve cells at the day 7 timepoint when the inflammatory phase is usually resolved (Figure 5B). Interestingly, the number of CD31 positive endothelial cells (Figure 5C) in the allylamine treated group appeared to be elevated compared to the acrylic acid and uncoated groups, however this trend did not reach significance by day 7.” We have now also expanded the discussion (lines 225-234) to suggest how the different dressings may be used in clinical applications, noting that this is not tested in the current study.

Reviewer 2 Report

The layout of the plasma polymerization setup is an essential part of this study. For that reason a more detailed description/drawing should be given in the text (not only a reference).

Estimated values for the thickness of the plasma polymerized layers should be given in the result section. Any impact of the coating on the surface morphology of the original wound dressing product should be mentioned and discussed.

The stability of the plasma polymerized layers in a wound dressing environment should be discussed. In the best case a proof should be provided.

Every abbreviation should be explained when it appears in the text for the first time.

For values that were determined experimentally, only significant digits should be given in the manuscript (line 119, 121).

The text should be double-checked for typing mistakes and minor language problems. In particular check wording in line 68-70 and 94-97.

Author Response

The layout of the plasma polymerization setup is an essential part of this study. For that reason a more detailed description/drawing should be given in the text (not only a reference).

Thank you for the suggestion. We have now included a schematic of the plasma set up in the materials and methods (Figure 6, Section 4.1 page 9, lines 250-253) which provides further detail regarding the plasma setup.

Estimated values for the thickness of the plasma polymerized layers should be given in the result section. Any impact of the coating on the surface morphology of the original wound dressing product should be mentioned and discussed. The stability of the plasma polymerized layers in a wound dressing environment should be discussed. In the best case a proof should be provided.

Our previous published work (Smith et al. ACS Appl. Mater. Interfaces 2016) has shown that these deposited acrylic acid and allylamine plasma polymer films have average thicknesses of 26.84nm and 30.01nm measured by atomic force microscopy. When placed into an aqueous environment the acrylic acid film largely dissolves and leaves a highly negatively charged acid film approximately 5nm thick. Conversely the allylamine film swells slightly producing a neutrally charged film with an average thickness of 31nm. Additionally both the acrylic acid and allylamine films had similar roughness’s when dry and deposited on smooth surfaces i.e. silicon wafer to enable accurate measurement (RMS (Root Mean Square) = 0.34 and 0.31 respectively). The film roughness increased when wet (RMS = 0.48 and 0.91 respectively). It is important to note that these measurements were taken after films were deposited onto silicon wafers. A hard, smooth substrate that allows for accurate measurements. Melolin is a cast polyester film and is therefore relatively soft and not smooth. Therefore, there will be subtle differences in the thickness of the films due to differences in the bonding of the plasma polymer to a polymeric substate rather than a ceramic / glass substrate. Additionally the inherent texture of the Melolin, which is visible to the naked eye and will therefore be of micron, even millimeter scale features will mask any changes in in the film thickness / roughness due to swelling / dissolution. This information has now been included in the results section 2.1, page 2, lines 73-90.

Every abbreviation should be explained when it appears in the text for the first time.

We apologise for the instances where an abbreviation was not explained. The manuscript has been revised and further information has been included for non-standard abbreviations XPS, RGB, MRP14 and CD31 at their first use.

For values that were determined experimentally, only significant digits should be given in the manuscript (line 119, 121).

We thank the reviewer for this advice. The results for the quantitative measurement of the Masson’s Trichrome staining in section 2.4  has been corrected to reflect the significant digits only. This can now be found on lines 149 and 151 of the revised manuscript.

The text should be double-checked for typing mistakes and minor language problems. In particular check wording in line 68-70 and 94-97.

We apologise for the lack of clarity in the text, particularly with these two identified sections. The resubmitted manuscript has been revised to improve the clarity of meaning.

Author Response

IJMS-2048800 Response to Reviewer 3

Thank you for your review of ijms-2048800-v2. We have addressed your comments below and uploaded the revised manuscript in line with the suggestions.

  1. Figure 2 and Section 3-Discussion: Please explain why the untreated Melolin has an accelerated wound closure between day 3 and day 5, and even has the lowest closure (around 55%), while those with plasma polymers are still in between 60-70% at day 5?

Mouse excisional wounds using this model of healing often display very little change in wound area during the first 3 days, after which the rate of closure increases (see Figure 1 SAL7-2 group from Sim et al IJMS 2022 https://www.mdpi.com/1422-0067/23/21/13655 and Figure 1 PBS control group from Thomas et al Reg Biomat 2021 https://academic.oup.com/rb/article/8/4/rbab024/6311484 ). As the untreated Melolin group represents the normal rate of healing, it is not that the untreated Melolin group has an accelerated wound closure between day 3 and 5, rather that there is a delay in the rate of healing between these days by both of the plasma functionalised dressings. As already presented and discussed in the manuscript, there is no significant difference in the amount of closure achieved by the allylamine functionalised dressing when compared to uncoated dressings on day 5, suggesting that it is only during the initial stages of wound healing that the stimulatory effect occurs. In section 2.3, we show the negative effect upon re-epithelialisation when treated with these dressing, which may result in the failure of the rapid healing that is normally observed between day 3 and 5. We have now expanded the discussion (line 202) to indicate that it may be the negative impact upon re-epithelisation by the allylamine functionalised dressings, that we believe has reduced the overall rate of healing. In contrast, the acrylic acid coated dressings, do not increase wound healing during the initial stages, which may be due to reduced migration of fibroblasts during the early stages of healing, which was unable to be overcome by and positive effect upon keratinocytes, but may have led to increased matrix deposition. This possibility is addressed further in the response to point 2 and also included in the discussion lines 210 - 216.

  1. Figure 2 and Section 3-Discussion: The significant difference of p p <0.05 for the wound closure at day 5 is between untreated Melolin (about 55%) and acrylic acid plasma polymerized Melolin (around 70%). So in this case (day 5), Melolin without any treatment performs better than those with plasma treatments. Why would that be?

As discussed in the manuscript, this was an unexpected result, as we had anticipated that the positive in vitro indications around this surface stimulating keratinocyte migration and proliferation would have translated to better wound re-epithelialisation. However, it may be that the dampening of fibroblast migration induced by this surface modification seen in vitro, may have resulted in a fibroblast phenotype predominantly facilitating ECM remodelling and collagen deposition. The Masson Trichrome results in Figure 4 and section 2.4 support this hypothesis, however further in depth analysis of this would be required in future studies to confirm this. We have added these points to the discussion in Section 3 lines 210 - 216.

  1. Figure 2 and Section 3-Discussion: It seems that the sample with plasma polymerized allylamine make the wound to be covered faster (day 3), but in the end (day 7) it shows similar result with that of the untreated Melolin and that of with acrylic acid plasma polymer (closing of around 35% from that of the initial wound area). What would happen if the same experiments are extended to 14 days? It would be interesting to see which one will have the smallest closure to, let’s say, 10%? To put it another way, why did you stop the experiment while the wound closure is still around 35%?

The endpoint of day 7 was chosen as this is usually the best day to demonstrate long lasting differences in healing. Should the initial ~20% improvement in healing seen in the allylamine functionalised dressing group have been sustained (i.e, had the apparent negative effect of this dressing upon reepithelialisation not occurred) we would have anticipated this group at the 7 day timepoint to be very close to complete healing (or the 10% closure suggested by the reviewer) and histological analysis would have identified significant improvements in the resolution of the wound. In our experience with this model, at day 14, 6mm excisional wounds are entirely healed and so differences in healing are difficult to discern as all achieve full closure.  As already discussed in Section 3, lines 179 – 206 it appears that the opposing effects of allylamine functionalisation to improve fibroblast migration but impede keratinocyte migration means that the initial positive effect of these dressings upon early wound closure, potentially though increased migration of fibroblasts into the wound site and more rapid provisional matrix deposition, is later offset through the prevention of keratinocyte migration and formation of a neo-epidermis, resulting in no difference by day 7.

  1. Section 1: There are only 24 references, which is not ideal. Please expand to > 30 references. The Section 1-Introduction (only two paragraphs, at this moment) critically needs expanded knowledge about plasma polymerization processes for biofunctionalization. Please refer to:
  2. Acrylic acid plasma polymers:
  • Acta Biomaterialia 11 (2015) 58-67 https://doi.org/10.1016/j.actbio.2014.09.027
  • Plasma Processes and Polymers 17 (2020) 1900209 https://doi.org/10.1002/ppap.201900209
  • Plasma Chemistry and Plasma Processing 41 (2021) 47-83 https://doi.org/10.1007/s11090-020-10135-6
  1. Allylamine plasma polymers:
  • Journal of Polymer Science Part B: Polymer Physics 51 (2013) 1361-1367 https://doi.org/10.1002/polb.23341
  • Biochemical Engineering Journal 78 (2013) 198-204 https://doi.org/10.1016/j.bej.2013.02.022
  • Plasma Processes and Polymers 12 (2015) 817-826 https://doi.org/10.1002/ppap.201400215
  1. Combination of acrylic acid and allylamine plasma polymers
  • Plasma Processes and Polymers 2 (2005) 641-649 https://doi.org/10.1002/ppap.200500043
  • Plasma Processes and Polymers 8 (2011) 208-214 https://doi.org/10.1002/ppap.201000111
  • Applied Surface Science 473 (2019) 838-847 https://doi.org/10.1016/j.apsusc.2018.12.216

Additional text and references have been added to the introduction and can be found on lines 44-61.

  1. Figure 1: Please explain why there is fluorine F1s peak after plasma polymerization treatment.

Fluorine is a common contamination in XPS spectra, especially in nitrogen containing films such as Allylamine. The contamination could have come from the XPS itself, multiuser instruments are commonly contaminated, it could also have come from the atmosphere post deposition. It cannot have come from the plasma reactor and therefore impacted deposition as the fluorine is present at such low levels, and also not present on the acrylic acid film.

  1. Figure 1: Please also make a table that contains all information in Figure 1a, 1b, 1c (binding energy, the related functional groups, percentage)

Thank you for the suggestion. A table summarising the elemental peak fits has now been included as supplementary material Table 1 and is referred to in the results section 2.1 on lines 86-87.

  1. Line 87: Please confirm which one is the intended meaning for “ceramic / glass”:
  • Is it “ceramic or glass substrate”, or
  • “composite of ceramic and glass”, or
  • “ceramic on top of glass” ?

Both are meant. Silicon wafers can be classed as a glass, ceramic or glass ceramic when discussing their materials categorisation. The primary point of this sentence is to highlight that the bonding of a plasma polymer film to a carbon rich polymer i.e. the Melolin used in this study and the bonding of the same plasma polymer film to an inorganic glass ceramic i.e. the silicon wafer used for XPS is anticipated to be different and therefore there may be subtle differences in the chemistry or structure of the films. This is clarified on lines 105.

  1. Line 89: Do you mean “thickness or roughness” for this line: “thickness / roughness”?

Both are meant. Changes in both the thickness of the films and the roughness of the films would be somewhat masked by the inherent texture of the melolin. The sentence on 107 has been reworded to clarify this.

  1. Line 89-90: Do you mean “swelling or dissolution” for this line: “swelling / dissolution”?

Both are meant, the acrylic acid film initially swells and then rapidly dissolves whilst the allylamine film just swells. The sentence on 108 has been reworded to clarify this.

  1. Line 13: 6 mm --> please add a space between the number and its unit

A space has been added between the number and its unit on line 13.

  1. Line 75-78: 26.84 nm…30.01 nm…5 mm…31 nm…--> please add a space between the number and its unit

A space has been added between the number and its unit on lines 93, 95 and 96.

  1. Line 118: n= 6 --> please add a space between the equal sign and a number.

A space has been added between the equal sign and number on lines 127, 141, 194 and 200.

  1. Figure 3 and its caption: Please make sure that a figure and its caption are in the same page.

Figure 3 and its caption are now on the same page.

  1. Line 128-132: …4 µm…500 µm…n= 6… --> please add a space between the number and its unit

Spaces have been added between the number and its unit on lines 136-139.

  1. Line 155: …MRP-14 positive --> do not use “+ve”

+ve has now been changed to positive on line 164.

  1. Line 156: CD31 positive --> do not use “+ve”

+ve has now been changed to positive on line 165.

  1. Line 158: …MRP-14 positive --> do not use “+ve”

+ve has now been changed to positive on line 167.

  1. Line 173: …Melolin… --> uppercase M, because it is a name or a brand.

An uppercase M has now been used on line 182.

  1. Line 177: re-epithelialisation --> please hyphenate, in order to be consistent with the rest of the manuscript.

Re-epithelialisation has now been hyphenated on line 186.

  1. Line 181: …4 µm…--> please add a space between the number and its unit

A space has been added between the number and its unit, now on line 191.

  1. Line 186-189: …4 µm …n= 6… --> please add a space between the number and its unit

Spaces have been added between the number and its unit on lines 196-200.

  1. Line 244-246: …10-2 ...10-2… --> superscripted -2

10-2 is now correctly presented on lines 259 and 261.

  1. Line 245-248: ..3 W…5 W…--> please add a space between the number and its unit

Spaces have been added between the number and its unit, now on lines 260-263.

  1. Line 274-277: …6 mm…0.5 cm…n= 6…1 cm × 1 cm...--> please add a space between the number and its unit

Spaces have been added between the number and its unit, now on lines 289-292.

  1. Line 277: …1 cm × 1 cm… --> please use the multiplication sign × instead of the lowercase letter x.

A multiplication sign instead of the lowercase letter x has now been used on line 292.

  1. Line 288: …The 4 µm… --> do not start a sentence with numbers, and please add a space between the number and its unit

Line 303 now begins with The 4 µm and a space added between the number and its unit.

  1. Line 289: …H&E (hematoxylin and eosin) staining… --> every abbreviation must be defined

The abbreviation H&E has now been defined on lines 304-305.

  1. Line 291: …4×… --> please use the multiplication sign × instead of the lowercase letter x, and the common sequence is the number first and then followed by the multiplication sign.

Line 306 now reads 4× magnification.

  1. Line 298: …10×… --> please use the multiplication sign × instead of the lowercase letter x, and the common sequence is the number first and then followed by the multiplication sign.

Line 314 now reads 10× magnification.

  1. Line 311-323: …100 µg/mL…100 µg/mL…250 mL…2 mg/mL…2 mg/mL…2 µg/mL…--> please add a space between the number and its unit, and please use uppercase L for mL instead of lowercase l (consistency issue).

Lines 328- 339 have all been corrected as suggested.

  1. Line 332: …A p-value… --> p-value is commonly hyphenated

The term p-value has now been hyphenated on line 348.

  1. Line 380: …cell adhesion.. --> please separate “cell” and “adhesion” with a space

A space has now been added between cell and adhesion on line 396.

Round 2

Reviewer 3 Report

Review of ijms-2048800-v3

Thank you for your effort in improving this manuscript, as well as for additional discussion and explanations. It can be accepted now.

Note: Line 112, caption of Figure 1, please remove "Biomaterials Science". It is enough to write "Robinson et al. [32]". This correction can be performed during the proofreading stage. Thanks.

Author Response

Thank you for the recommendation, "Biomaterials Science" has been removed from Line 112, caption of Figure 1.